# RETHINKING TRANSFORMER THROUGH DUAL BANACH SPACES

## ABSTRACT

Transformers have significantly advanced deep learning across multiple domains, yet the theoretical foundations with respect to their structure remain an open area of research. In this paper, we introduce a novel perspective by interpreting Transformers through the framework of dual Banach spaces. Specifically, we prove that the exponentiated query-key kernel in the attention mechanism can be interpreted as a bilinear form on Banach spaces. Building on this, we provide a theoretical proof demonstrating that the attention mechanism in Transformers can be viewed as a dual space operator, while feed-forward networks function as a correction mechanism between dual solution and primal solution. To demonstrate the benefits of the dual Banach space perspective, we show how this framework introduces a novel form of regularization for Transformer. These findings offer new insights into understanding and potentially improving Transformer architectures using principled mathematical frameworks.

## 1 INTRODUCTION

Transformers (Vaswani et al., 2017) have revolutionized deep learning across various domains, achieving state-of-the-art performance in natural language processing, computer vision, speech recognition, and multimodal learning (Devlin et al., 2019; Dosovitskiy et al.; Kim et al., 2024). At the core of the Transformer architecture lies the attention mechanism, followed by a feed-forward network, which facilitates efficient representation learning and contextual understanding. Additionally, Transformers are compatible with existing regularization techniques, such as weight regularization, enhancing their generalization capabilities.

Despite their empirical success, a comprehensive theoretical understanding of their functioning and optimization remains an open research question. Prior works have explored the theoretical foundations and properties of Transformers (Tsai et al., 2019; Edelman et al., 2022), offering valuable insights into their operations as interacting particle systems (Geshkovski et al., 2023) and their algorithmic reasoning capabilities (Sanford et al., 2024). However, the theoretical foundations of Transformer architectures remain an open area of research, such as optimization dynamics and the functional roles of their components in complex tasks (Elhage et al., 2021).

In this paper, we introduce a novel perspective by interpreting Transformers through the framework of dual Banach spaces. Unlike conventional analyses that rely on Euclidean spaces, this Banach space perspective offers a mathematically rigorous functional view that provides deeper insights into Transformer architectures. We establish that attention in Transformers can be interpreted as solving an optimization problem in a Banach space. Specifically, attention is a dual space operator on Banach spaces, and feed-forward networks is a correction mechanism between dual solution and primal solution.

This theoretical framework based on the dual Banach space perspective allows us to develop functional regularization techniques that leverage the structural properties of Transformers, resulting in improved generalization and smoother decision boundaries. We conduct experiments across vision and language tasks, demonstrating that dual Banach regularization can enhance performance of Transformer models. These findings offer new insights into understanding and potentially improving Transformer architectures using principled mathematical frameworks.

Our main contributions are as follows:

- We prove that the exponentiated query-key kernel in the attention mechanism can be interpreted as a bilinear form on a Banach space.

- We interpret the attention operation as an operator in a dual Banach space and the feed-forward network as a correction mechanism between dual solution and primal solution.
- We propose a new functional regularization based on the dual Banach space framework, which improves generalization with smooth decision boundaries across vision and language tasks.

## 2 PRELIMINARIES

### 2.1 REGULAR BANACH SPACE

Let $(\Omega, \mathcal{F}, \mu)$ be a complete $\sigma$-finite measure space and let $\mathcal{L}^0(\Omega, \mathcal{F}, \mu)$ be the space of all equivalence classes of $\mu$-measurable real-valued functions endowed with the topology of convergence in measure relative to each set of finite measure. A normed vector space $\mathcal{B}$ is called a **Banach space** if its norm induces a complete metric, or more precisely, every Cauchy sequence of $\mathcal{B}$ is convergent.

A **bilinear form** between two Banach spaces $\mathcal{B}_1, \mathcal{B}_2$ is a function $\langle \cdot, \cdot \rangle_{\mathcal{B}_1 \times \mathcal{B}_2}$ from $\mathcal{B}_1 \times \mathcal{B}_2$ to $\Re$ that is linear about both arguments. It is said to be **continuous** if there exists a positive constant $C$ such that for all $f \in \mathcal{B}_1, g \in \mathcal{B}_2$

$$|\langle f, g \rangle_{\mathcal{B}_1 \times \mathcal{B}_2}| \le C \|f\|_{\mathcal{B}_1} \|g\|_{\mathcal{B}_2}$$

Note that for notational convenience, we write $\langle v_1, v_2 \rangle_{V_1 \times V_2} = \langle v_2, v_1 \rangle_{V_2 \times V_1}$, , but this does not imply symmetry, which requires $\langle v_1, v_2 \rangle_{V_1 \times V_2} = \langle v_2, v_1 \rangle_{V_1 \times V_2}$.

A Banach space $\mathcal{B}$ is called a **regular Banach space (RBS)** on $\Omega$ if for each $f \in \mathcal{B}$ (a function on $\Omega$), its norm $\|f\|_{\mathcal{B}} = 0$ if and only if $f = 0$ everywhere on $\Omega$ and every point evaluation functional $\delta_x, x \in \Omega$ on $\mathcal{B}$ is continuous, that is, there exists a $C_x > 0$ for all $f \in \mathcal{B}$

$$|\delta_x(f)| = |f(x)| \le C_x \|f\|_{\mathcal{B}}$$

When $\mathcal{B}$ is an RBS on $\Omega$, and $f, f_n \in \mathcal{B}$, then $\|f_n - f\|_{\mathcal{B}}$ implies $f_n(x) \to f(x)$, as $n \to \infty$, for each $x \in \Omega$. Moreover, the point-evaluation functionals $\delta_x(f) \in \mathcal{B}^*$ for all $x \in \Omega$ by the definition of RBS.

### 2.2 DUAL BANACH SPACE

In functional analysis, a **dual space** $\mathcal{B}^*$ refers to the space of all continuous (bounded) linear functionals on a Banach space $\mathcal{B}$ and becomes a Banach space when it is endowed with the norm, say

$$\|\nu\|_{\mathcal{B}^*} := \sup_{f \in \mathcal{B}, f \neq 0} \frac{\nu(f)}{\|f\|_{\mathcal{B}}}, \quad \forall \nu \in \mathcal{B}^*$$

The **dual bilinear form** $\langle \cdot, \cdot \rangle_{\mathcal{B}}$ is defined on the Banach space $\mathcal{B}$ and its dual space $\mathcal{B}^*$ as

$$\langle f, \nu \rangle_{\mathcal{B}} := \langle f, \nu \rangle_{\mathcal{B} \times \mathcal{B}^*} = \nu(f), f \in \mathcal{B}, \nu \in \mathcal{B}^*.$$

A specific example of an element (operator) in the dual space $\mathcal{B}^*$ is the partial derivative operator evaluated at a point $\boldsymbol{x} \in \mathbb{R}^d$, denoted by $\nu_{\boldsymbol{x}}^j := \frac{\partial}{\partial x_j}\big|_{\boldsymbol{x}}$. This operator satisfies

$$\langle f, \nu_{\boldsymbol{x}}^j \rangle_{\mathcal{B} \times \mathcal{B}^*} = \frac{\partial}{\partial x_j} f(\boldsymbol{x}), \ f \in \mathcal{B}, \nu_{\boldsymbol{x}}^j \in \mathcal{B}^*.$$

where the pairing $\langle \cdot, \cdot \rangle_{\mathcal{B} \times \mathcal{B}^*}$ evaluates the partial derivative of $f$ at $\boldsymbol{x}$.

In the Reproducing Kernel Hilbert Spaces (RKHS) framework, a Hilbert space is isometrically isomorphic to its dual space, which results in a unique reproducing kernel for the RKHS. An RBS is, however, not isometrically isomorphic to its dual space, and the continuity of point-evaluation functionals does not guarantee the existence of a kernel.

We call $\mathcal{G}(f) := \partial \|\cdot\|_{\mathcal{B}}(f)$ Gâteaux derivative of $\|\cdot\|_{\mathcal{B}}$ at $f \in \mathcal{B}$ (by defining $\mathcal{G}(f) = 0$ if $f = 0$) if for all $f \in \mathcal{B} \setminus \{0\}, g \in \mathcal{B}$, there exists a continuous linear functional, denoted by $\mathcal{G}(f) \in \mathcal{B}^*$, such that

$$\langle g, \mathcal{G}(f) \rangle_{\mathcal{B}} := \lim_{\lambda \to 0} \frac{\|f + \lambda g\|_{\mathcal{B}} - \|f\|_{\mathcal{B}}}{\lambda} \tag{1}$$

exists (and uniformly converge). It follows from (1) that $|\langle g, \mathcal{G}(f) \rangle_{\mathcal{B}}| \leq \|g\|_{\mathcal{B}}$ and $\langle f, \mathcal{G}(f) \rangle_{\mathcal{B}} = \|f\|_{\mathcal{B}}$, which leads to $\|\mathcal{G}(f)\|_{\mathcal{B}^*} = 1$. Also,

$$\frac{\nu}{\|\nu\|_{\mathcal{B}^*}} \in \partial \| \cdot \|_{\mathcal{B}}(f) \quad \text{if and only if} \quad \frac{f}{\|f\|_{\mathcal{B}}} \in \partial \| \cdot \|_{\mathcal{B}^*}(\nu) \tag{2}$$

We call an RBS $\mathcal{B}$ **reflexive** if $(\mathcal{B}^*)^* = \mathcal{B}$, that is, any continuous linear functional $T$ on $V^*$ must be of the form:

$$T(v^*) = \langle u, v^* \rangle_V, v^* \in V^*$$

for some $u \in V$.

# 3 FUNCTIONAL ANALYSIS OF TRANSFORMER IN DUAL BANACH SPACES

In this section, we establish the theoretical connection between Transformer architectures and dual Banach spaces. We first prove how the exponentiated query-key kernel in the attention mechanism can be interpreted as a bilinear form on Banach spaces (Section 3.1). We then demonstrate that the attention operation can be formally represented as dual space operators (Section 3.2). Finally, we analyze the feed-forward network as a correction mechanism that address the discrepancy between dual representations and desired primal solutions (Section 3.3).

## 3.1 EXPONENTIATED QUERY-KEY KERNEL AS A BILINEAR FORM ON BANACH SPACES

According to (Megginson, 2012), we define the **annihilators** of a set $A \in V_1$ in $V_2$ and a set $B \in V_2$ in $V_1$ with respect to $\langle \cdot, \cdot \rangle_{V_1 \times V_2}$ by

$$(A^{\perp})_{V_1 \times V_2} := \{b \in V_2 : \langle a, b \rangle_{V_1 \times V_2} = 0, \ \forall a \in \text{span}(A)\} \subset V_2$$

$$(^{\perp}B)_{V_1 \times V_2} := \{a \in V_1 : \langle a, b \rangle_{V_1 \times V_2} = 0, \ \forall b \in \text{span}(B)\} \subset V_1$$

**Theorem 3.1.** *(A pair of feature maps) Let $\mathcal{H}_1, \mathcal{H}_2$ be two distinct Hilbert spaces of the same cardinality with a continuous bilinear form $\langle \cdot, \cdot \rangle_{\mathcal{H}_1 \times \mathcal{H}_2}$. Suppose that there exists $\Phi_1 : \Omega_1 \to \mathcal{H}_1$, and $\Phi_2 : \Omega_2 \to \mathcal{H}_2$ such that the following **null conditions** are satisfied:*

$$(\Phi_1(\Omega_1)^{\perp})_{\mathcal{H}_1 \times \mathcal{H}_2} = \{0\} \quad \text{and} \quad (^{\perp}\Phi_2(\Omega_2))_{\mathcal{H}_1 \times \mathcal{H}_2} = \{0\} \tag{3}$$

*Let us define $\mathcal{B}_1$ and $\mathcal{B}_2$ by*

$$\mathcal{B}_1 := \{f_w : \Omega_1 \to \mathbb{R} \mid f_w(x) = \langle \Phi_1(\cdot), w \rangle_{\mathcal{H}_1 \times \mathcal{H}_2}, \ w \in \mathcal{H}_2, \ x \in \Omega_1\}$$

$$\mathcal{B}_2 := \{g_v : \Omega_2 \to \mathbb{R} \mid g_v(y) = \langle v, \Phi_2(y) \rangle_{\mathcal{H}_1 \times \mathcal{H}_2}, \ v \in \mathcal{H}_1, \ y \in \Omega_2\}$$

*with norm $\|f_w\|_{\mathcal{B}_1} = \|w\|_{\mathcal{H}_2}$ and $\|g_v\|_{\mathcal{B}_2} = \|v\|_{\mathcal{H}_1}$.*

*Then, $\mathcal{B}_1$ and $\mathcal{B}_2$ are regular Banach spaces (RBS). Also the bilinear form on $\mathcal{B}_1 \times \mathcal{B}_2$ defined by*

$$\langle f_w, g_v \rangle_{\mathcal{B}_1 \times \mathcal{B}_2} = \langle v, w \rangle_{\mathcal{H}_1 \times \mathcal{H}_2}, \quad \forall v \in \mathcal{H}_1, \forall w \in \mathcal{H}_2$$

*is a continuous bilinear form.*

*Proof.* From the null condition (3), $f_w$ and $g_v$ are both unique and well-defined. Following the derivations in (Lin et al., 2022), we have for all $f_w \in \mathcal{B}_1$,

$$|f_w(x)| = |\langle \Phi_1(x), w \rangle_{\mathcal{H}_1 \times \mathcal{H}_2}| \leq C \|\Phi_1(x)\|_{\mathcal{H}_1} \|w\|_{\mathcal{H}_2} = C \|\Phi_1(x)\|_{\mathcal{H}_1} \|f_w\|_{\mathcal{B}_1} \quad \forall x \in \Omega_1$$

and similarly for all $g_v \in \mathcal{B}_2$. Hence, point evaluation functionals are continuous on both $\mathcal{B}_1$ and $\mathcal{B}_2$ and therefore $\mathcal{B}_1$ and $\mathcal{B}_2$ are RBS. Also from

$$|\langle f_w, g_v \rangle_{\mathcal{B}_1 \times \mathcal{B}_2}| = |\langle v, w \rangle_{\mathcal{H}_1 \times \mathcal{H}_2}| \leq C \|v\|_{\mathcal{H}_1} \|w\|_{\mathcal{H}_2} = C \|f_w\|_{\mathcal{B}_1} \|g_v\|_{\mathcal{B}_2}$$

$\langle \cdot, \cdot \rangle_{\mathcal{B}_1 \times \mathcal{B}_2}$ is a continuous bilinear form.
Now for all $x \in \Omega_1, f_w \in \mathcal{B}_1$,

$$f_w(x) = \langle \Phi_1(x), w \rangle_{\mathcal{H}_1 \times \mathcal{H}_2} = \langle f_w, g_{\Phi_1(x)} \rangle_{\mathcal{B}_1 \times \mathcal{B}_2}$$

and similarly for all $y \in \Omega_2, g_v \in \mathcal{B}_2$,

$$g_v(y) = \langle v, \Phi_2(y) \rangle_{\mathcal{H}_1 \times \mathcal{H}_2} = \langle f_{\Phi_2(y)}, g_v \rangle_{\mathcal{B}_1 \times \mathcal{B}_2}$$

$\square$

It is necessary for the Hilbert spaces $\mathcal{H}_1$ and $\mathcal{H}_2$ to have the same cardinality, as the null conditions may otherwise fail to hold. In this theorem, we refer to the mappings $\Phi_1 : \Omega_1 \to \mathcal{H}_1$, and $\Phi_2 : \Omega_2 \to \mathcal{H}_2$ as **a pair of feature maps** and $\mathcal{H}_1$ and $\mathcal{H}_2$ as a pair of feature spaces. This construction preserves the desirable properties of the Reproducing Kernel Banach Spaces (RKBS) framework while relaxing its structural assumptions, thereby offering greater modeling flexibility.

For instance, consider Hilbert spaces $\mathcal{H}_1 = \mathbb{R}^m$ and $\mathcal{H}_2 = \mathbb{R}^n$, endowed with a weighted dot product defined as

$$\langle \mathbf{v}, \mathbf{w} \rangle_{\mathcal{H}_1 \times \mathcal{H}_2} = \mathbf{v}^T \mathbf{G} \mathbf{w}, \quad \mathbf{v} \in \mathbb{R}^m, \ \mathbf{w} \in \mathbb{R}^n$$

where $\mathbf{G} \in \mathbb{R}^{m \times n}$ is an arbitrary (possibly indefinite) matrix.

In standard Transformers, attention is computed through the inner product of linearly projected representations, expressed as $\langle W_q \hat{\boldsymbol{x}}_i^l, W_k \hat{\boldsymbol{x}}_j^l \rangle$. However, since the layer immediately preceding the attention mechanism typically ends with a shared non-linear feed-forward network (FFN), applying different linear projections afterward is effectively equivalent to applying two distinct non-linear transformations. Consequently, attention operates on non-linear feature representations, which can be interpreted as $\langle \Phi_1, \Phi_2 \rangle$. This observation motivates the use of a pair of feature maps in our formulation.

**Lemma 1.** *(Exponentiated query-key kernel as a bilinear form on Banach spaces) Let us denote the exponentiated query-key kernel used in the (scaled) dot-product attention mechanism,*

$$\exp(\boldsymbol{q}_i^T \boldsymbol{k}_j / \sqrt{d_k}), \tag{4}$$

*where $\boldsymbol{q}_i = W_q \boldsymbol{x}_i + \boldsymbol{b}_q \in \mathbb{R}^d, \boldsymbol{k}_j = W_k \boldsymbol{y}_j + \boldsymbol{b}_k \in \mathbb{R}^d, W_q \in \mathbb{R}^{d \times d_s}, \boldsymbol{x}_i \in \mathbb{R}^{d_s}, \boldsymbol{b}_q \in \mathbb{R}^d, W_k \in \mathbb{R}^{d \times d_t}, \boldsymbol{y}_j \in \mathbb{R}^{d_t}, $ and $\boldsymbol{b}_k \in \mathbb{R}^d$. Given Theorem 3.1, (4) can be represented as an inner product $\langle \Phi_1(\boldsymbol{x}_i), \Phi_2(\boldsymbol{x}_j) \rangle_{\mathcal{H}_1 \times \mathcal{H}_2}$ for suitable feature maps $\Phi_1$ and $\Phi_2$ into appropriately chosen Hilbert spaces.*

*Proof.* See Appendix B.1. $\square$

## 3.2 ATTENTION AS DUAL SPACE OPERATOR

Given a fixed finite set of distinct points $\boldsymbol{x}_j \in \Re^D$ with $\mathbf{y}^\tau = (y_j^\tau)_{j=1,\dots,n} \in \Re^n$, for $j = 1, \dots, n$ and $\tau \in \mathcal{T} = \{1, \dots, D\}$, the regularization problem is given by

$$\inf\{ \mathcal{L}(\mathbf{y}, \mathcal{V}(f_\tau)) + \lambda \varphi(\|f_\tau\|_{\mathcal{B}}) : f_\tau \in \mathcal{B} \} \tag{5}$$

where an operator $\mathcal{V} : \mathcal{B} \to \Re^n$ is defined by $\mathcal{V}(f_\tau) = (\langle f_\tau, \nu_j \rangle_{\mathcal{B}})_{j=1,\dots,n}$ for all $f_\tau \in \mathcal{B}$ with (linearly independent) **dual space operators** $\nu_j \in \mathcal{B}^*$, e.g., $\langle f_\tau, \nu_j \rangle = f_\tau(\boldsymbol{x}_j)$. Note that $\nu_j = \delta_{x_j}$ when $\mathcal{B}$ is a Hilbert space. $\mathcal{L}(\mathbf{y}^\tau, \cdot) : \Re^n \to \Re^+, \varphi : \Re^+ \to \Re^+$ is a *strictly increasing* smooth regularizer and $\lambda$ is a positive regularization parameter.

To guarantee the existence of well-defined dual solutions in this generalized setting, we assume that $\mathcal{B}$ is a Banach space whose dual $\mathcal{B}^*$ is smooth, and that the set of dual space operators $\nu_j \in \mathcal{B}^*$ for $j = 1, \dots, N$ is linearly independent. Then, in accordance with our definitions, the Represener theorems (Wang & Xu, 2021) for regularized learning in Banach spaces can be formalized as follows:

**Theorem 3.2.** *(Represener theorem in Banach spaces (Wang & Xu, 2021)) Let $\hat{f}_\tau \in \mathcal{B}$ be a primal solution of the regularization problem (5), then the following holds for $\tau \in \mathcal{T}$:*

*The dual solution $\hat{f}_\tau^* \in \mathcal{B}^*$ is a linear functional given by*

$$\hat{f}_\tau^*(\cdot) := \sum_{j=1}^{n} c_j^\tau \nu_j \in \mathcal{B}^*, \ \exists \ c_j^\tau \in \Re, j = 1, \dots, n. \tag{6}$$

*The primal solution $f$ can be reconstructed as*

$$\hat{f}_\tau = \gamma_\tau \mathcal{G}^*(\hat{f}_\tau^*) \quad where \quad \langle \hat{f}_\tau, \hat{f}_\tau^* \rangle_{\mathcal{B}} = \|\hat{f}_\tau\|_{\mathcal{B}} \|\hat{f}_\tau^*\|_{\mathcal{B}^*}. \tag{7}$$

In contrast, the dual representation $\hat{f}_\tau^*$ in Theorem 3.2 operates on functions rather than on individual points. As a result, it cannot be directly interpreted as a function mapping points to values. This motivates the representation of

attention as a dual operator, highlighting a key distinction from traditional asymmetric kernel representations in the RKBS framework.

The central challenge is recovering the primal solution from its dual representation. As shown in the next theorem, this dual formulation exhibits a strong resemblance to the attention mechanism used in Transformer models. We proceed to demonstrate how attention in Transformers can be formally interpreted through the framework of dual Banach space operators.

**Theorem 3.3.** *(Attention as a dual space operator) The dual solution $\hat{f}_\tau^*$ can be expressed in the following form:*

$$\hat{f}_\tau^*(\cdot) = \sum_{j=1}^n c_j^\tau \langle \Phi_1(\boldsymbol{x}_j), \Phi_2(\cdot) \rangle_{\mathcal{B}_1 \times \mathcal{B}_2} \tag{8}$$

*where $\Phi_1$ and $\Phi_2$ are suitably chosen feature maps into Hilbert spaces $\mathcal{H}_1$ and $\mathcal{H}_2$, respectively. Notably, the operator $\Phi_2(\boldsymbol{x})$ operates on functions—specifically on $\Phi_1(\boldsymbol{x}_j)$, which itself is a function—rather than directly on individual points. Specifically, the output of the attention block takes the form:*

$$\tilde{f}_\tau^*(\boldsymbol{x}_i^l) = \sum_{j=1}^n \underbrace{(W_o W_v \hat{\boldsymbol{x}}_j^l)}_{\tilde{c}_j^\tau} \times \underbrace{\text{softmax}\left( \frac{W_q \hat{\boldsymbol{x}}_i^l \cdot (W_k \hat{\boldsymbol{x}}_j^l)}{\sqrt{d_k}} \right)}_{\langle \Phi_1(\hat{\boldsymbol{x}}_j^l), \Phi_2(\boldsymbol{x}_i^l) \rangle_{\mathcal{B}_1 \times \mathcal{B}_2}} \tag{9}$$

*Proof.* See Appendix B.2. □

This theorem highlights the functional similarity between the dual representation in Banach spaces and the attention mechanism in Transformer models. Specifically, the learned query-key interactions correspond to a bilinear pairing between non-linear feature maps, while the value projection yields the linearly adapted coefficients $\tilde{c}_j^\tau$. This structure naturally aligns with the attention mechanism, which can be viewed as a realization $\tilde{f}_\tau^*$ of the dual solution $\hat{f}_\tau^*$ in the primal space.

It is important to note that the attention mechanism arises from a specific choice of the feature map pair $(\Phi_1, \Phi_2)$. Alternative choices of such mappings can lead to entirely different mechanisms, thereby offering greater flexibility in the design of novel algorithms within the dual Banach space framework. Although these alternative duality mappings may increase the complexity of reconstructing the primal function $f$ from its dual representation $f^*$, demonstrating new directions for algorithmic design beyond the standard attention mechanism. In addition, this can be easily extended to multi-head attention, which is presented in Appendix.

### 3.3 Feed-forward Network as a Correction Mechanism

Our framework interprets the Transformer architecture through the lens of dual Banach spaces. From the perspective of duality, recovering the primal solution $f \in \mathcal{B}$ from its dual representation $f^* \in \mathcal{B}^*$ is a nontrivial problem. This is fundamentally different from the Reproducing Kernel Hilbert Space (RKHS) framework, where a space is isometrically isomorphic to its dual, simplifying the relationship between primal and dual forms. In the more general Banach space context, this relationship is governed by a highly non-linear duality map.

As defined in our preliminaries, the Gâteaux derivative of the norm, $\mathcal{G} : \mathcal{B} \to \mathcal{B}^*$, serves as this duality map. It links an element $f$ in the primal space to a corresponding functional $\mathcal{G}(f)$ in the dual space. Consequently, the task of reconstructing the primal solution $\hat{f}_\tau$ from a dual solution $\hat{f}_\tau^*$ (approximated by the attention mechanism) is equivalent to computing the inverse of the duality map, i.e., $\hat{f}_\tau \propto \mathcal{G}^{-1}(\hat{f}_\tau^*)$.

Computing this inverse map, $\mathcal{G}^{-1}$, is generally intractable as it lacks a closed-form expression in non-Hilbert Banach spaces. This necessitates the use of iterative numerical methods to find a solution. Fixed-point proximity algorithms (Li et al., 2019) are a class of such methods designed for complex, non-differentiable optimization problems. These algorithms repeatedly apply certain operators (e.g., projections or soft-thresholding) to a variable until it converges to a fixed point, which corresponds to the desired primal solution. However, applying this theoretical machinery directly to Transformers is infeasible. The underlying function spaces are extremely high-dimensional, and the specific forms of

the proximity operators, which may be defined through partial differential equations, are unknown and incomputable for functions represented by deep neural networks.

To bridge this theoretical gap, we hypothesize that the feed-forward network (FFN) sublayer learns to approximate a single iteration of such a fixed-point recovery algorithm. FFN acts as a data-driven operator that learns a local approximation of the inverse duality map, correcting the dual representation from the attention layer to better align with the desired primal solution. This interpretation provides a concrete mechanism for the structure of Transformer block, where attention approximates the dual solution and the FFN performs a corrective step towards the primal solution:

$$\hat{f}_\tau(x) \approx \tilde{f}_\tau^*(x) + \text{FFN}(\text{LayerNorm}(\tilde{f}_\tau^*(x))) \quad \text{where} \quad \text{FFN}(z) = W_2\sigma(W_1 z). \tag{10}$$

This formulation can be viewed as one step of a learned fixed-point iteration, $v_{k+1} \approx v_k + \Delta_k$, where the initial state $v_k$ is the attention output $\tilde{f}_\tau^*(x)$ and the update $\Delta_k$, which approximates the action of $\mathcal{G}^{-1}$, is provided by the FFN.

We have empirically validated this hypothesis in a controlled experiment, demonstrating that a network analogous to a FFN can successfully learn a theoretically grounded fixed-point proximity step. This provides strong evidence that the FFN functions as an effective, learnable corrector. The full details and results of this experiment are provided in Appendix D.

Our dual Banach space perspective yields a principled explanation for the interaction between attention and feed-forward layers. Specifically, the architecture can be viewed as a two-step operator: first approximating a dual representation, then applying a learned proximity correction to recover the primal solution. This interpretation provides a mathematical link between its components and fundamental operator-theoretic constructs.

## 4 DUAL BANACH SPACE PERSPECTIVE ON REGULARIZING TRANSFORMER

In this section, we provide an example of the benefits offered by the dual Banach space perspective: a novel form of regularization for Transformers. We begin by revisiting Equation (10). The mathematical analysis in (10) shows that the approximated dual space representation, $\tilde{f}_\tau^*$, can be transformed into a more accurate solution in the primal space, $\hat{f}_\tau$. Since both terms are in the same Banach space, $\mathcal{B}$, we can reformulate the Banach space norm as follows:

$$\|\hat{f}_\tau\|_\mathcal{B} = \|\tilde{f}_\tau^* + \text{FFN}(\text{LayerNorm}(\tilde{f}_\tau^*))\|_\mathcal{B}. \tag{11}$$

Given the $\ell^2$ norm in the embedding space (Shawe-Taylor & Cristianini, 2004), the following proposition holds under the commonly satisfied relaxed conditions.

**Proposition 4.1.** *(Norm Bound for Single Layer Transformer) Let $\tilde{f}_\tau^*$ be the approximated dual solution computed by the attention mechanism as defined in (9) and (10). Also, $\hat{f}_\tau = \tilde{f}_\tau^* + \text{FFN}(\text{LayerNorm}(\tilde{f}_\tau^*))$ be the corresponding primal space solution of (5). Then, given an input $\boldsymbol{x}$, the norm of $\hat{f}_\tau$ is bounded by:*

$$\|\hat{f}_\tau\|_\mathcal{B} \leq \alpha \cdot \left(1 + \|W_2\| \cdot \|W_1\| \cdot \frac{\max|\gamma|}{\text{std}(\tilde{f}_\tau^*(\boldsymbol{x}))}\right) \cdot \|\tilde{f}_\tau^*(\boldsymbol{x})\|_{\ell^2}, \tag{12}$$

*where $\alpha > 0$ and $\gamma$ is a scale parameter in* LayerNorm.

*Proof.* See Appendix B.3. $\square$

### 4.1 DUAL BANACH REGULARIZATION WITH THE PRACTICAL NORM BOUND

Based on Proposition 4.1, we derive an empirical training framework for Transformers. By selecting the strictly increasing regularizer $\varphi(x) = x^2$ as in Equation 5, we extend Proposition 4.1 to multi-layer Transformers with $L$ layers. The resulting framework is expressed as follows:

$$\mathcal{L}(\tilde{f}_\tau^*(\boldsymbol{x}), \boldsymbol{y}) + \lambda \cdot \frac{1}{L}\sum_{l=1}^{L}\left(1 + \|W_2^l\| \cdot \|W_1^l\| \cdot \frac{\max|\gamma^l|}{\text{std}(\tilde{f}_\tau^{*,l}(\boldsymbol{x}^l))}\right)^2 \cdot \|\tilde{f}_\tau^{*,l}(\boldsymbol{x}^l)\|_{\ell^2}^2, \tag{13}$$

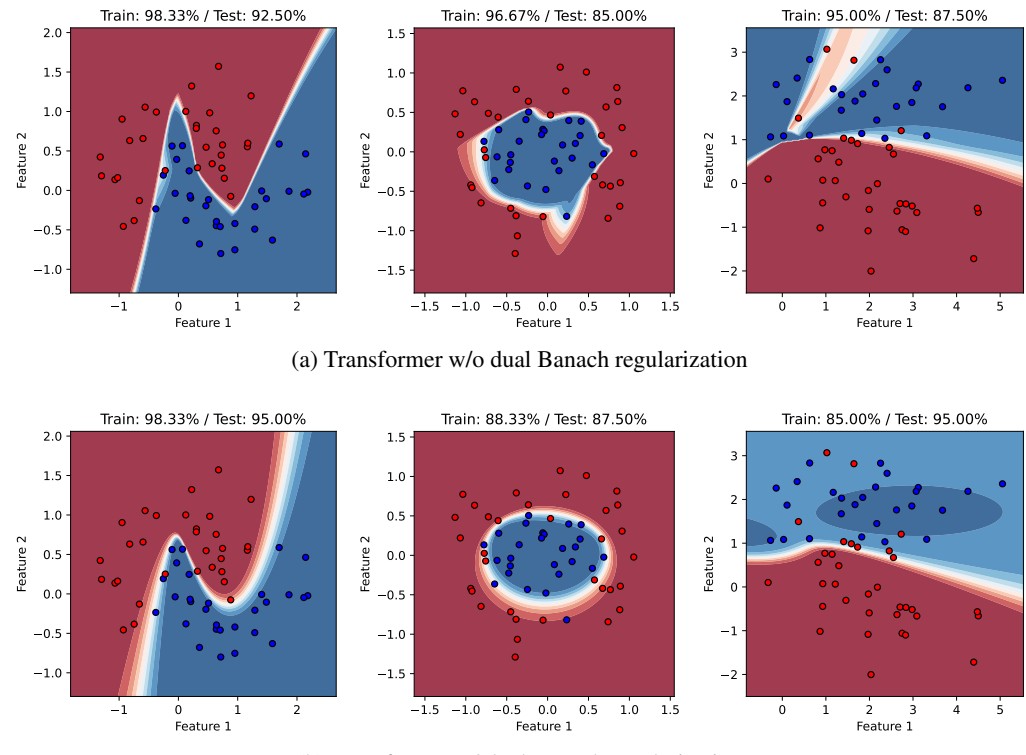

(a) Transformer w/o dual Banach regularization

(b) Transformer w/ dual Banach regularization

Figure 1: **Effect of dual Banach regularization.** Visualization of decision boundaries and train/test accuracies of Transformers trained without (top row) and with (bottom row) dual Banach regularization across three synthetic datasets. Without regularization, the models exhibit highly distorted decision boundaries and lower test accuracy. In contrast, dual Banach regularization results in smoother and more stable decision boundaries, improving generalization ability.

where $\lambda$ is a hyperparameter that controls the balance between the task loss $\mathcal{L}(\tilde{f}_\tau^*(\boldsymbol{x}), \boldsymbol{y})$ and norm bound loss. This framework can be easily extended to Transformers with multi-head attention, which can be found in Appendix. We refer to this regularization framework as **Dual Banach (DB) Regularization.**

With the proposed dual Banach regularization, we conduct an experiment on three two-dimensional synthetic datasets with added noise to test generalization. Detailed experimental settings are provided in Appendix F.1. As illustrated in Figure 1, the Transformer model trained without regularization tends to overfit, resulting in highly distorted decision boundaries and lower test accuracies. In contrast, the model trained with DB regularization exhibits smoother decision boundaries that better capture the underlying data structure, leading to significantly higher test accuracies. These results demonstrate that DB regularization acts as an effective inductive bias, enabling the model to avoid overfitting and learn more meaningful patterns from noisy inputs.

## 4.2 Empirical Verification of dual Banach regularization

In this subsection, we empirically evaluate the dual Banach regularization approach across multiple domains to validate its practical utility. Our experiments are specifically designed to address scenarios where large Transformer models are applied to small datasets, which typically leads to overfitting.

**Image Classification**  To evaluate the effectiveness of dual Banach regularization in image classification tasks, we conducted experiments on the CIFAR-10/100 datasets and their corrupted counterparts, CIFAR-10-C/100-C (Hendrycks & Dietterich, 2019). Following the experimental setup of Guo et al. (2023), we used Vision Transformer (ViT) models (Dosovitskiy et al., 2020). As summarized in Table 1, applying $\mathcal{L}_{\text{DB}}$ consistently improves performance and robustness across all settings. For instance, on the ViT-S model, our method boosts the average accuracy on CIFAR-10-C by **3.08%**

Table 1: Image classification accuracy (%) on CIFAR-10 and CIFAR-100, and their corrupted versions, CIFAR-10-C and CIFAR-100-C. We also provide individual results for three noises: Guassian noise, shot noise, and pixelate. Avg. indicates the overall accuracy on CIFAR-$\star$-C. Results are presented as mean $\pm$ confidence interval, with **bold** numbers indicating the best performance. **Blue** indicates the gap between Baseline.

| CIFAR-10 and CIFAR-10-C | | | | | |
|---|---|---|---|---|---|
| **Method** | **No Corruption** | **Gaussian Noise** | **Shot Noise** | **Pixelate** | **Avg.** |
| ViT-S | 95.04±0.85 | 79.81±2.22 | 83.68±1.68 | 86.56±1.97 | 88.47±1.32 |
| $+ \mathcal{L}_{DB}$ | **96.80±0.20** (+1.76) | **85.38±0.37** (+5.57) | **88.51±0.34** (+4.83) | **90.88±0.65** (+4.32) | **91.55±0.18** (+3.08) |
| ViT-T | 94.01±0.24 | 77.84±3.86 | 81.75±2.38 | 86.14±1.30 | 87.15±0.25 |
| $+ \mathcal{L}_{DB}$ | **95.74±0.18** (+1.73) | **81.76±0.67** (+3.92) | **85.31±0.50** (+3.56) | **88.27±1.14** (+2.13) | **89.53±0.13** (+2.38) |
| **CIFAR-100 and CIFAR-100-C** | | | | | |
| **Method** | **No Corruption** | **Gaussian Noise** | **Shot Noise** | **Pixelate** | **Avg.** |
| ViT-S | 74.03±1.52 | 49.42±3.10 | 54.79±2.81 | 63.76±3.20 | 62.49±1.79 |
| $+ \mathcal{L}_{DB}$ | **78.85±0.43** (+4.82) | **57.79±1.11** (+8.37) | **62.92±1.19** (+8.13) | **71.18±0.73** (+7.42) | **68.43±0.43** (+5.94) |
| ViT-T | 72.93±0.41 | 49.04±0.52 | 54.39±0.72 | 61.96±0.17 | 61.68±0.29 |
| $+ \mathcal{L}_{DB}$ | **77.01±1.12** (+4.08) | **53.75±2.92** (+4.71) | **58.76±2.06** (+4.37) | **68.15±2.17** (+6.19) | **65.89±1.28** (+4.21) |

and on CIFAR-100-C by **5.94%**. Furthermore, the regularization enhances training stability, evidenced by significantly narrower confidence intervals. Notably, the regularized ViT-T achieves performance comparable to the larger ViT-S baseline, highlighting the effectiveness of our approach in promoting parameter-efficient learning.

To further evaluate our approach, we conducted experiments on the CUB-200-2011 fine-grained classification dataset (Wah et al., 2011). We extracted features using a pre-trained CLIP ViT-B/32 (Radford et al., 2021) and then trained a simple Transformer classifier with 2 tokens, 1 layer, and 512 embedding dimensions. To quantify decision boundary smoothness, we employed the widely-used inverse-margin (IM) (Pitas et al., 2017; Jiang et al.), where a lower IM value corresponds to a smoother decision boundary. As the strength of the Dual Banach regularization increases, as shown

Table 2: Effectiveness of Dual Banach regularization on CUB-200-2011.

| Method | Accuracy (%) | IM ($\downarrow$) |
|---|---|---|
| Baseline | 71.99 | 29.84 |
| $+ \mathcal{L}_{DB}(\lambda = 0.1)$ | 74.39 | 21.27 |
| $+ \mathcal{L}_{DB}(\lambda = 0.5)$ | 75.51 | 17.43 |
| $+ \mathcal{L}_{DB}(\lambda = 1.0)$ | 75.46 | 14.22 |

in Table 2, we observe a consistent improvement in test accuracy. Moreover, the IM value steadily decreases, which empirically demonstrates that our regularization method leads to a smoother decision boundary and better generalization.

**Natural Language Processing** We extend our evaluation to natural language processing using decoder-only transformer architectures with causal attention. This experiment is particularly important as it verifies that our theoretical framework applies not only to bidirectional attention mechanisms but also to causal attention settings commonly used in generative language models. We use GPT-2 (Radford et al., 2019) on the WikiText-103 benchmark dataset (Merity et al., 2017), which consists of over 100 million tokens extracted from high-quality Wikipedia articles. For our experiments, we train models with and without dual Banach regularization to assess the impact on language modeling performance. We use the AdamW-schedulefree optimizer (Defazio et al., 2024), which recently achieved the state-of-the-art performance on optimization by alleviating the difficulty of learning rate scheduling.

The experimental results on language modeling are summarized in Table 3, which presents the performance of GPT-2 under different dual Banach regularization strengths. Our results demonstrate that increasing the regularization strength ($\lambda$) consistently improves model performance, with the highest regularization value ($\lambda = 5.0$) achieving the best results in both evaluation loss ($3.02 \pm 0.01$) and perplexity ($20.49 \pm 0.10$). While low regularization ($\lambda = 0.5$) showed some improvements in consistency as evidenced by its narrower confidence intervals, the differences in performance compared to higher regularization strengths were notable. The results suggest that stronger dual Banach regularization values may provide more substantial benefits for language modeling tasks.

As shown in Fig. 2, models with our regularization exhibit different training dynamics. The perplexity curves show that the model with higher regularization (especially, $\lambda = 5.0$) maintains a lower perplexity than that of the baseline ($\lambda = 0$)

Table 3: Effectiveness of dual Banach regularization on GPT-2 on WikiText-103. Results are presented as mean $\pm$ confidence interval, with **bold** numbers indicating the best performance. **Blue** indicates the gap between Baseline.

| Method | Perplexity ($\downarrow$) |
|---|---|
| Baseline | 21.09$\pm$0.10 |
| + $\mathcal{L}_{DB}(\lambda = 0.5)$ | 20.83$\pm$0.01 (0.26) |
| + $\mathcal{L}_{DB}(\lambda = 1.0)$ | 20.74$\pm$0.12 (0.35) |
| + $\mathcal{L}_{DB}(\lambda = 5.0)$ | **20.49$\pm$0.10 (0.60)** |

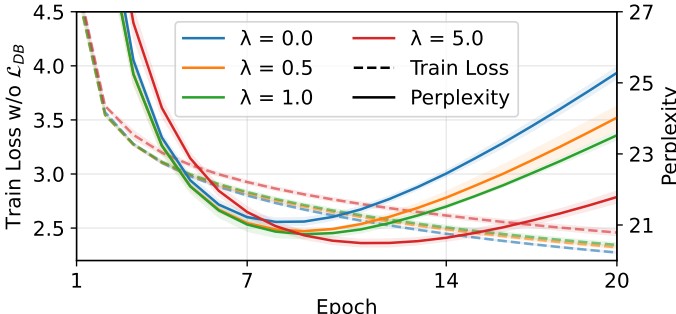

Figure 2: Training loss (dashed) and perplexity (solid). The shaded areas indicate 95% confidence intervals.

Table 4: Performance comparison of varying weight decay strengths with and without Dual Banach regularization ($\mathcal{L}_{DB}$). For ViT-S, we report average accuracy (%) on CIFAR-10 / CIFAR-10-C. For GPT-2, we report perplexity ($\downarrow$) on WikiText-103. The results demonstrate that $\mathcal{L}_{DB}$ provides consistent improvements across all settings.

| | ViT-S (CIFAR-10 / 10-C Acc. $\uparrow$) | | | GPT-2 (Perplexity $\downarrow$) | | | |
|---|---|---|---|---|---|---|---|
| Weight Decay | No Reg | + $\mathcal{L}_{DB}$ | Improvement | $\lambda = 0.0$ | $\lambda = 0.5$ | $\lambda = 1.0$ | $\lambda = 5.0$ |
| 0.00 | 94.22 / 87.21 | 96.49 / 90.88 | +2.27 / +3.67 | 21.82 | 21.55 | 21.36 | 21.16 |
| 0.05 | 95.04 / 88.47 | 96.80 / 91.55 | +1.76 / +3.08 | 21.09 | 20.83 | 20.74 | 20.49 |
| 0.10 | 95.79 / 89.59 | 96.88 / 91.77 | +1.09 / +2.18 | 20.43 | 20.15 | 20.08 | 20.01 |

throughout training. As training epoch increases after epoch 10, the gap between perplexity increases. This tells us that our dual Banach regularization effectively improves the generalization ability of language models.

### 4.3 COMPLEMENTARITY WITH EXISTING REGULARIZATION TECHNIQUES

Our regularizer is complementary to, rather than a replacement for, existing regularization techniques. We conduct additional experiments to evaluate its interaction with two widely-used methods: weight decay and word dropout. We first explored the complementary effects between $\mathcal{L}_{DB}$ and weight decay of AdamW. As shown in Table 4, our regularizer provides consistent performance gains when combined with various weight decay strengths. For ViT-S, $\mathcal{L}_{DB}$ improves accuracy on CIFAR-10-C across all tested weight decay values. Correspondingly, for GPT-2 on WikiText-103, our method consistently lowers perplexity. Notably, the best results are often achieved through the joint application of both techniques, which suggests that $\mathcal{L}_{DB}$ addresses model complexity in a manner distinct from simple parameter norm penalties.

We further investigate the integration of our method with word dropout, an input-level regularization technique. We trained GPT-2 on WikiText-103 with a word dropout probability of 0.1. The results in Table 5 demonstrate that our regularizer improves perplexity beyond word dropout. This supports our hypothesis that the functional regularization from our framework is orthog-

Table 5: Effectiveness of $\mathcal{L}_{DB}$ with word dropout ($p = 0.1$) on WikiText-103 (Perplexity $\downarrow$).

| Method | $\lambda = 0.0$ | $\lambda = 0.5$ | $\lambda = 1.0$ | $\lambda = 5.0$ |
|---|---|---|---|---|
| Baseline | 21.09 | 20.83 | 20.74 | 20.49 |
| + Word Dropout | 19.66 | 19.41 | 19.26 | 19.12 |

onal to input-level stochasticity, enabling enhanced generalization when applied jointly. These experiments validate that our regularizer can be seamlessly integrated into existing training pipelines as a complementary component to further increase model performance and robustness.

### 5 CONCLUSION

In this paper, we introduce a theoretical framework analyzing Transformer architectures via dual Banach spaces. By modeling attention as solutions to a regularization problem and feed-forward networks as corrections between dual solutions and attention, our approach enables the development of functional regularization techniques to improve generalization. We hope this insight inspire new Transformer designs with stronger theoretical guarantees in future work.

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

## A   LIMITATION AND DISCUSSION

Our proposed dual Banach space framework provides theoretical insights and introduces effective regularization for Transformers. However, its applicability to large-scale models, such as recent large language models, remains an open research question due to our computational constraints. Additionally, our approach primarily targets regularization of the entire Transformer during full training, which limits its direct use in fine-tuning scenarios where only a subset of parameters is updated. In future work, we will explore adaptive regularization strategies that dynamically adjust during training and develop computationally efficient variants suitable for fine-tuning.

## B   PROOFS

### B.1   PROOF OF LEMMA 1

*Proof.* First, the inner product $\boldsymbol{q}_i^T \boldsymbol{k}_j$ is itself a valid kernel. It is also well known that if $p(x)$ is a polynomial with positive coefficients and $k(x, y)$ is a kernel, then $p(k(x, y))$ is also a kernel (Shawe-Taylor & Cristianini, 2004). The exponential function can be uniformly approximated by such polynomials (i.e., its Taylor series expansion). Therefore, the exponential of a kernel can be viewed as a pointwise limit of kernels.

Because the property of finite positive semi-definiteness is preserved under pointwise limits, it follows that

$$\exp(\boldsymbol{q}_i^T \boldsymbol{k}_j / \sqrt{d_k})$$

is itself a valid kernel. See (Wright & Gonzalez, 2021) for another specific example of such feature map representations.
$\square$

### B.2   PROOF OF THEOREM 3.3

*Proof.* It follows directly from Theorem 3.2 that the dual solution $\hat{f}_\tau^*$ admits the following representation:

$$\hat{f}_\tau^*(\cdot) = \sum_{j=1}^{n} c_j^\tau \langle \Phi_1(\mathbf{x}_j), \Phi_2(\cdot) \rangle_{\mathcal{H}_1 \times \mathcal{H}_2} \tag{14}$$

where $\Phi_1$ and $\Phi_2$ are appropriately defined feature maps into Hilbert spaces $\mathcal{H}_1$ and $\mathcal{H}_2$, respectively. Importantly, the mapping $\Phi_2(\boldsymbol{x})$ acts on functions—specifically on $\Phi_1(\boldsymbol{x}_j)$, which itself is a function—rather than directly on individual points.

Given a linear differential operator $\mathbf{D}$, there exists a uniquely determined adjoint operator, denoted by $\tilde{\mathbf{D}}$, such that for any sufficiently differentiable functions $u(\mathbf{x})$ and $v(\mathbf{x})$ satisfying appropriate boundary conditions, the following identity holds:

$$\langle u, \mathbf{D}v \rangle = \langle \tilde{\mathbf{D}}u, v \rangle$$

as established in (Lanczos, 1996).

Now, define the feature map $\phi_1(\mathbf{x}_j)$ by

$$\phi_1(\mathbf{x}_j)(\cdot) = \exp\left(-\frac{1}{2\sigma_i^2} \| \cdot - \mathbf{x}_j \|^2 \right)$$

which corresponds to a Gaussian radial basis function centered at $\mathbf{x}_j$.

Let $\mathbf{L}$ denote a linear differential operator. Its associated self-adjoint operator, along with the operator $\mathbf{D}$ and its adjoint $\tilde{\mathbf{D}}$, are given respectively by:

$$\mathbf{L} = \tilde{\mathbf{D}}\mathbf{D} = \sum_{n=0}^{\infty} (-1)^n \alpha_n \nabla^{2n}, \qquad \alpha_n = \frac{\sigma_i^{2n}}{n! 2^n}$$

$$\mathbf{D} = \sum_{a+b+\cdots+k=n} \alpha_n^{1/2} \frac{\partial^n}{\partial x_1^a \partial x_2^b \cdots \partial x_d^k},$$

$$\tilde{\mathbf{D}} = \sum_{a+b+\cdots+k=n} (-1)^n \alpha_n^{1/2} \frac{\partial^n}{\partial x_1^a \partial x_2^b \cdots \partial x_d^k}$$

where $\nabla^{2n}$ is the iterated Laplacian operator given by

$$\nabla^{2n} = \frac{\partial^2}{\partial x_1^2} + \frac{\partial^2}{\partial x_2^2} + \cdots + \frac{\partial^2}{\partial x_d^2}$$

Here, $\mathbf{D}$ is a generalized differential operator expressed as a sum over all multi-index partial derivatives of total order $n$, with $\tilde{\mathbf{D}}$ defined as its formal adjoint under integration by parts, assuming appropriate boundary conditions. The self-adjoint operator $\mathbf{L}$ is then constructed as the composition $\tilde{\mathbf{D}}\mathbf{D}$, resulting in an infinite series of even-order Laplacian-type operators weighted by $\alpha_n$.

Then, under appropriate conditions, we have (Poggio & Girosi, 1990)

$$\langle \phi_1(\mathbf{x}_j), \mathbf{L} \rangle_{\mathcal{H}_1 \times \mathcal{H}_2} = \delta_{\mathbf{x}_j}$$

Now define the dual space operator $\phi_2(\cdot)$ by $\phi_2(\mathbf{x}_i) = \langle \phi_1(\mathbf{x}_i), \mathbf{L} \rangle_{\mathcal{H}_1 \times \mathcal{H}_1}$. Then the following holds:

$$\langle \phi_1(\mathbf{x}_j), \phi_2(\mathbf{x}_i) \rangle_{\mathcal{B}_1 \times \mathcal{B}_2} = \phi_1(\mathbf{x}_j)(\mathbf{x}_i) = \exp\left( -\frac{1}{2\sigma_i^2} \|\mathbf{x}_j - \mathbf{x}_i\|^2 \right)$$

Observe that the Gaussian kernel can also be expressed as a normalized inner product in exponential form:

$$\exp\left( \frac{\langle \mathbf{x}_j, \mathbf{x}_i \rangle}{\sigma_i^2} \right) = \frac{\exp\left( -\frac{1}{2\sigma_i^2} \|\mathbf{x}_j - \mathbf{x}_i\|^2 \right)}{\sqrt{\exp\left( -\frac{1}{\sigma_i^2} \|\mathbf{x}_j\|^2 \right) \exp\left( -\frac{1}{\sigma_i^2} \|\mathbf{x}_i\|^2 \right)}} = \langle \phi_1(\mathbf{x}_j), \phi_1(\mathbf{x}_i) \rangle_{\mathcal{H}_1 \times \mathcal{H}_1}$$

By normalizing the feature maps as $\tilde{\phi}_1 = \phi_1 / \|\phi_1\|_{\mathcal{H}_1}$ and $\tilde{\phi}_2 = \phi_2 / \|\phi_2\|_{\mathcal{H}_2}$, we obtain:

$$\langle \tilde{\phi}_1(\mathbf{x}_j), \tilde{\phi}_2(\mathbf{x}_i) \rangle = \exp\left( \frac{\langle \mathbf{x}_j, \mathbf{x}_i \rangle}{\sigma_i^2} \right)$$

A similar result can also be obtained by defining the feature map

$$\hat{\phi}_1(\mathbf{x}_j) = \exp\left( \frac{\langle \mathbf{x}_j, \cdot \rangle}{\sigma_i^2} \right)$$

and setting $\hat{\phi}_2(\mathbf{x}_i) = \langle \phi_1(\mathbf{x}i), \mathbf{L} \rangle \mathcal{H}_1 \times \mathcal{H}_1$. With these definitions, we again recover the inner product form

$$\langle \hat{\phi}_1(\mathbf{x}_j), \hat{\phi}_2(\mathbf{x}_i) \rangle = \exp\left( \frac{\langle \mathbf{x}_j, \mathbf{x}_i \rangle}{\sigma_i^2} \right)$$

These constructions naturally lead to a generalization of the attention mechanism in Transformer architectures. Specifically, the inner product between suitably chosen feature maps $\Phi_1$ and $\Phi_2$ can be expressed as

$$\langle \Phi_1(\hat{\boldsymbol{x}}_j^l), \Phi_2(\hat{\boldsymbol{x}}_i^l) \rangle_{\mathcal{B}_1 \times \mathcal{B}_2} = \mathrm{softmax}\left( \frac{W_q \hat{\boldsymbol{x}}_i^l \cdot (W_k \hat{\boldsymbol{x}}_j^l)}{\sqrt{d_k}} \right)$$

This formulation reinforces the interpretation of the attention score as a bilinear pairing between distinct feature representations in dual Banach spaces. It is worth noting that $c_j^\tau$ typically represents a nonlinear transformation of $\hat{\boldsymbol{x}}_j^l$; however, in our approach, the attention mechanism employs a linearly adapted variant, $\tilde{c}_j^\tau$. $\square$

### B.3 PROOF OF PROPOSITION 4.1

*Proof.* By the representation theorem stated in Lemma 1, the solution $\hat{f}_\tau$ lies in a finite-dimensional subspace of $\mathcal{B}$. Therefore, by the norm equivalence theorem for finite-dimensional normed spaces, there exist constants $C_1, C_2 > 0$ such that:

$$C_1 \|\hat{f}_\tau\|_{\ell^2} \leq \|\hat{f}_\tau\|_{\mathcal{B}} \leq C_2 \|\hat{f}_\tau\|_{\ell^2} \tag{15}$$

By the definition of $\hat{f}_\tau$, we have:

$$\hat{f}_\tau = \tilde{f}_\tau^* + \mathrm{FFN}(\mathrm{LN}(\tilde{f}_\tau^*)) \tag{16}$$

Applying the triangle inequality for the $\ell^2$ norm:

$$\|\hat{f}_\tau\|_{\ell^2} \le \|\tilde{f}_\tau^*\|_{\ell^2} + \|\text{FFN}(\text{LN}(\tilde{f}_\tau^*))\|_{\ell^2} \tag{17}$$

For the FFN component, which takes the form $\text{FFN}(z) = W_2\sigma(W_1 z)$, we can apply the submultiplicative property of the norm:

$$\|\text{FFN}(\text{LN}(\tilde{f}_\tau^*))\|_{\ell^2} \le \|W_2\| \cdot \|W_1\| \cdot \|\text{LN}(\tilde{f}_\tau^*)\|_{\ell^2} \tag{18}$$

Let $\mathbf{x} \in \mathbb{R}^d$ with mean $\mu(\mathbf{x}) = \frac{1}{d}\sum_{i=1}^d x_i$ and variance $\sigma(\mathbf{x})^2 = \frac{1}{d}\|\mathbf{x} - \mu(\mathbf{x})\mathbf{1}\|_2^2$, where $\mathbf{1} = (1,\dots,1)^\top \in \mathbb{R}^d$. For simplicity, we only consider a multiplication parameter $\gamma$. Then, the LayerNorm operation can be formalized as:

$$\text{LN}(\mathbf{x}) = \gamma \odot \frac{\mathbf{x} - \mu(\mathbf{x})\mathbf{1}}{\sqrt{\sigma(\mathbf{x})^2 + \varepsilon}} \quad \in \mathbb{R}^d, \tag{19}$$

where $\gamma \in \mathbb{R}^d$, $\odot$ denotes elementwise multiplication, and $\varepsilon > 0$ is a small constant.

The mean-centering operation can be expressed as $\mathbf{x} - \mu(\mathbf{x})\mathbf{1} = P\mathbf{x}$, where $P = I_d - \frac{1}{d}\mathbf{1}\mathbf{1}^\top$ is an orthogonal projector with $\|P\|_2 = 1$. Then, LN in matrix form for norm analysis:

$$\text{LN}(\mathbf{x}) = \text{diag}(\gamma) \cdot \frac{1}{\sqrt{\sigma(\mathbf{x})^2 + \varepsilon}} \cdot P\mathbf{x} \tag{20}$$

Since $\|\text{diag}(\gamma)\|_2 = \max_{1 \le i \le d} |\gamma_i|$ and $\|P\|_2 = 1$, we have:

$$\|\text{LN}\|_{\ell^2} \le \max_i |\gamma_i| \cdot \frac{1}{\sqrt{\sigma(\mathbf{x})^2 + \varepsilon}} \tag{21}$$

Under practical conditions where $\sqrt{\sigma(\mathbf{x})^2 + \varepsilon} \approx \text{std}(\tilde{f}_\tau^*)$ remains bounded away from zero, we obtain the approximation:

$$\|\text{LN}(\tilde{f}_\tau^*)\|_{\ell^2} \approx \frac{\max|\gamma|}{\text{std}(\tilde{f}_\tau^*)} \cdot \|\tilde{f}_\tau^*\|_{\ell^2} \tag{22}$$

By combining these inequalities, we have

$$\|\hat{f}_\tau\|_{\ell^2} \le \|\tilde{f}_\tau^*\|_{\ell^2} + \|W_2\| \cdot \|W_1\| \cdot \frac{\max|\gamma|}{\text{std}(\tilde{f}_\tau^*)} \cdot \|\tilde{f}_\tau^*\|_{\ell^2} \tag{23}$$

$$= \left(1 + \|W_2\| \cdot \|W_1\| \cdot \frac{\max|\gamma|}{\text{std}(\tilde{f}_\tau^*)}\right) \cdot \|\tilde{f}_\tau^*\|_{\ell^2} \tag{24}$$

Therefore,

$$\|\hat{f}_\tau\|_{\mathcal{B}} \le C_2\|\hat{f}_\tau\|_{\ell^2} \le C_2 \cdot \left(1 + \|W_2\| \cdot \|W_1\| \cdot \frac{\max|\gamma|}{\text{std}(\tilde{f}_\tau^*)}\right) \cdot \|\tilde{f}_\tau^*\|_{\ell^2} \tag{25}$$

For the case with $\beta$, which is a shifting parameter in Layernorm, the term $\|W_2\|\cdot\|W_1\|\cdot\|\beta\|$ is added to the upper bound, which is a general weight regularization term.

This demonstrates that controlling the $\ell^2$ norm of $\tilde{f}_\tau^*$ effectively bounds the Banach space norm of $\hat{f}_\tau$ in the regularization problem 5. $\qquad\square$

## C   MULTI-HEAD ATTENTION AS DUAL SPACE OPERATOR

Multi-head attention extends the single-head mechanism by dividing the embedding dimension $D$ into $H$ heads, each of width $d_h \approx D/H$. The standard implementation applies separate projections for each head and subsequently concatenates the results before applying a final output projection.

For each head $h = 1,\dots,H$, projection matrices are defined as:

$$W_{q,h},\, W_{k,h},\, W_{v,h} :\ \mathbb{R}^D \ \to\ \mathbb{R}^{d_h}. \tag{26}$$

Similar to the single-head case, the output projection matrix $W_{o,h} : \mathbb{R}^{d_h} \to \mathbb{R}^D$ for each head $h$ benefits from the linearity property. This allows us to apply $W_{o,h}$ to each transformed value vector $W_{v,h}\hat{\boldsymbol{x}}_j^l$ individually before weighting by attention scores. Therefore, the multi-head attention block can be expressed as:

$$\tilde{f}_\tau^*(\boldsymbol{x}_i^l) = \sum_{h=1}^H \sum_{j=1}^n \underbrace{(W_{o,h}W_{v,h}\hat{\boldsymbol{x}}_j^l)}_{\tilde{c}_{j,h}^\tau} \times \underbrace{\mathrm{softmax}\left(\frac{(W_{q,h}\hat{\boldsymbol{x}}_i^l) \cdot (W_{k,h}\hat{\boldsymbol{x}}_j^l)}{\sqrt{d_h}}\right)}_{\langle \Phi_1(W_{k,h}\hat{\boldsymbol{x}}_j^l), \Phi_2(\boldsymbol{x}_i^l)\rangle_{\mathcal{B}_1 \times \mathcal{B}_2}} \tag{27}$$

Thus, multi-head attention can be interpreted as a dual-space operator characterized by multiple bilinear forms.

## D  EMPIRICAL VALIDATION OF THE FFN AS A CORRECTION MECHANISM

To empirically validate our hypothesis that the Feed-forward Network (FFN) acts as a learned proximity corrector, we conducted a controlled experiment on a problem where a theoretically-grounded oracle algorithm exists. The goal was to test whether a simple MLP, analogous to an FFN, could learn to approximate a single, corrective step of a fixed-point proximity algorithm.

### D.1  EXPERIMENTAL SETUP

We follow the setup of (Li et al., 2019) for a regularized $\ell_1$-SVM problem. The task is to find a solution that minimizes a combination of a data fidelity term (hinge loss) and a regularization term ($\ell_1$-norm). This non-differentiable optimization problem can be solved effectively using a two-step fixed-point proximity algorithm.

**Problem Formulation**  The state variable for the algorithm is a concatenated vector $v \triangleq [w; y]^T$, where $w$ contains the model weights and $y$ is an auxiliary variable. The algorithm seeks a fixed point of an operator $T$, such that $v = T(E(v))$, where $E$ and $T$ characterize the problem's structure. The deviation from this condition is measured by the fixed-point residual, defined as $r(v) \triangleq v - T(E(v))$.

**Oracle Algorithm**  The two-step proximity algorithm from (Li et al., 2019) serves as our oracle. For the $\ell_1$-SVM problem, the iterative update steps are given by:

$$w_{k+1} = \mathrm{prox}_{\frac{1}{\lambda}\phi}\left(w_k - \frac{C\beta}{\lambda}B^T(y_k - y_{k-1})\right) \tag{28}$$

$$y_{k+1} = \left(I - \mathrm{prox}_{\frac{1}{C\beta}\psi}\right)(y_k + Bw_{k+1}) \tag{29}$$

where $v_k = [w_k; y_k]^T$ is the state at iteration $k$. The matrix $B$ is constructed from the training data $(X, \boldsymbol{y}_{\mathrm{labels}})$ as $B = \mathrm{diag}(\boldsymbol{y}_{\mathrm{labels}})[X, \mathbf{1}]$. The operators $\mathrm{prox}_\phi$ and $\mathrm{prox}_\psi$ are the proximity operators corresponding to the $\ell_1$-regularizer and the hinge loss, respectively. For a function $g$, its proximity operator is defined as $\mathrm{prox}_g(z) = \mathrm{argmin}_u\left(g(u) + \frac{1}{2}\|u - z\|^2\right)$. For our specific problem, they have the following closed-form solutions:

- **$\ell_1$-norm (Soft-thresholding):** $(\mathrm{prox}_{\frac{1}{\lambda}\phi}(z))_j = \max\{|z_j| - \frac{1}{\lambda}, 0\} \cdot \mathrm{sgn}(z_j)$.

- **Hinge Loss:** $(\mathrm{prox}_{\frac{1}{C\beta}\psi}(z))_j$ is the solution to $\mathrm{argmin}_{x_j \in \mathbb{R}}\left\{\frac{1}{2}(x_j - z_j)^2 + \frac{1}{C\beta}(1 - x_j)_+\right\}$.

**Learned Corrector**  The oracle's update from $v_k$ to $v_{k+1}$ defines a reference correction step $\Delta_k^{\mathrm{ref}} \triangleq v_{k+1} - v_k$. Our experiment tests if a neural network can learn this correction. We train a 3-layer MLP (ReLU, width 1024) to predict the correction based on the previous two states: $\hat{\Delta}_k = \mathrm{FFN}(v_k, v_{k-1})$. The network's performance is then evaluated by applying its predicted correction, $v_{k+1}^{\mathrm{FFN}} = v_k + \hat{\Delta}_k$.

**Data and Parameters**  We generate a synthetic dataset of $n = 64$ samples in $d = 10$ dimensions from a standard Gaussian distribution $X \sim \mathcal{N}(0, I)$, with labels $y_{\mathrm{labels}} = \mathrm{sign}(X_{:,0})$. The algorithm's hyperparameters are set to $C = 1$, $\beta = 1$, and $\lambda = 1$.

## D.2 RESULTS AND DISCUSSION

We evaluated the learned FFN corrector against the oracle two-step proximity algorithm across several metrics. The results, summarized in Table 6, demonstrate that the FFN successfully learns the underlying mathematical structure of the correction step.

Table 6: Performance comparison between the learned FFN corrector and the oracle Two-Step Proximity Algorithm. The FFN closely mimics the behavior and effectiveness of the theoretically derived oracle.

| Metric | Definition | FFN Corrector | Two-Step Prox. Algorithm (Oracle) |
|---|---|---|---|
| Median Residual Ratio | $\|r(v_{k+1})\|/\|r(v_k)\|$ | 0.266 | 0.210 |
| Objective-Gap Reduction | $L(v_k) - L(v_{k+1})$ | 0.021 | 0.015 |
| Alignment with Oracle | $\langle \hat{\Delta}_k, \Delta_k^{\mathrm{ref}} \rangle / \|\Delta_k^{\mathrm{ref}}\|^2$ | 0.996 | - |
| Iterations to Converge | Num. Iterations to tolerance | 741 | 756 |

The most remarkable result is the Alignment, which measures the cosine similarity between the direction of the predicted correction of FFN ($\hat{\Delta}_k$) and true correction of the oracle ($\Delta_k^{\mathrm{ref}}$). An alignment of 0.996 indicates that the FFN learns a correction step that is almost perfectly aligned with the theoretically optimal direction. Furthermore, the FFN effectively reduces the fixed-point residual and the objective-function gap, and even converges slightly faster than the oracle on average.

These results provide strong empirical support for central claim of our paper : the FFN in a Transformer is not merely a generic function approximator but serves the specific role of a **data-driven residual corrector**, learning to perform a step of a fixed-point algorithm to map an approximated dual representation back towards its corresponding primal solution.

# E COMPUTATIONAL EFFICIENCY OF DUAL BANACH REGULARIZATION

## E.1 EXPERIMENTAL ANALYSIS

Our dual Banach regularization is designed to be computationally efficient and highly scalable, making it practical for use with large-scale Transformer models. The regularization term introduces only a modest overhead in terms of both Floating Point Operations (FLOPs) and memory usage during training.

As summarized in Table 7, the computational overhead is minimal and, notably, its relative impact decreases as the model size increases. For instance, the FLOPs overhead for the ViT-S model is 6.2%, while for the larger GPT-2 model, it drops to just 2.0%. This demonstrates the favorable scaling properties of our approach.

Table 7: Computational overhead of dual Banach regularization. The overhead is modest and scales favorably, with the relative cost decreasing for larger models like GPT-2.

| Model | FLOPs (G) | | | Memory (MiB) | | |
|---|---|---|---|---|---|---|
| | Baseline | $+ \mathcal{L}_{\mathrm{DB}}$ | Overhead (%) | Baseline | $+ \mathcal{L}_{\mathrm{DB}}$ | Overhead (%) |
| ViT-T/16 | 1.25 | 1.38 | 10.4% | 4691 | 5600 | 19.4% |
| ViT-S/16 | 4.60 | 4.89 | 6.2% | 9125 | 10948 | 20.0% |
| GPT-2 | 9.53 | 9.72 | 2.0% | 13542 | 14698 | 8.5% |

The primary reason for this efficiency is the computational complexity of the regularizer. While the standard self-attention mechanism has a computational cost that scales quadratically with the sequence length $T$ (i.e., $\mathcal{O}(T^2)$), the cost of our dual Banach regularizer scales only linearly with the sequence length (i.e., $\mathcal{O}(T)$). This linear scaling ensures that for modern Transformers with very long contexts, the relative cost of our regularization becomes increasingly negligible compared to the dominant cost of the attention computation. This makes $\mathcal{L}_{\mathrm{DB}}$ a practical and scalable solution for improving the performance and robustness of large foundation models without incurring significant computational penalties.

## E.2 DETAILED ANALYSIS OF COMPUTATIONAL COST

We provide a detailed breakdown of the computational cost for a standard Transformer block and our proposed dual Banach regularization term, $\mathcal{L}_{\text{DB}}$. To ensure consistency with the original analysis, the following calculations are presented on a per-instance basis (i.e., batch size $B = 1$) and use the 1 FLOP/MAC (multiply-add) convention.

**Notation** Let $T$ be the sequence length, $C$ be the embedding dimension, $H$ be the number of attention heads, $D = C/H$ be the dimension per head, and $r$ be the MLP expansion ratio.

**Baseline Transformer FLOPs** The FLOPs for a standard Transformer block are the sum of the self-attention and FFN costs. The total cost per layer is approximately:

$$\text{FLOPs}_{\text{Total}} \approx \underbrace{2T^2C}_{\text{Attention Matmuls}} + \underbrace{4TC^2}_{\text{Projections}} + \underbrace{2rTC^2}_{\text{FFN}}.$$

**Dual Banach Regularizer FLOPs** The additional cost from $\mathcal{L}_{\text{DB}}$ is substantially lower and scales more favorably. It arises from two main terms, with costs identical to our original analysis:

- **Attention Output Norm** ($\|\tilde{f}_\tau^*(\cdot)\|_{\ell_2}^2$)**:** The first term computes the squared $\ell_2$-norm of the attention output. The total FLOPs for this term per layer are:

$$\text{FLOPs}_{\text{Term 1}} \approx HD^2C + THD^2 + THD + TC.$$

- **FFN and LayerNorm Term:** The second term involves norms of the FFN weight matrices and LayerNorm statistics. The cost is:

$$\text{FLOPs}_{\text{Term 2}} \approx \frac{3}{2}TC + 3rC^2 + \frac{3}{2}(r+2)C.$$

The total computational cost of our regularizer is dominated by terms that scale as $\mathcal{O}(T)$. In contrast, the baseline self-attention mechanism contains terms that scale quadratically as $\mathcal{O}(T^2)$. Therefore, as sequence length $T$ increases—a key trend in modern LLMs—the relative overhead of our $\mathcal{O}(T)$ regularizer becomes insignificant compared to the $\mathcal{O}(T^2)$ cost of the attention block, confirming its efficiency and scalability.

## F EXPERIMENTAL SETUP AND ADDITIONAL RESULTS

### F.1 EXPERIMENTAL SETUP OF TOY EXPERIMENT

The experiments were conducted on widely used two-dimensional synthetic datasets, including (i) interleaving half-moon structures, (ii) concentric circular decision boundaries, and (iii) normally distributed point clusters that are linearly separable.[1] To evaluate generalization performance, we artificially added noise to the input features. The data were split into training and test sets with a 6:4 ratio using a fixed random seed of 42. We constructed a simple Transformer model with an input dimension of 2, a single encoder layer, 4 attention heads, and hidden dimensions of 16 for both the attention and feed-forward networks. We applied layer normalization and trained the models to classify samples into two classes using the Adam optimizer with a learning rate of 0.01 for 500 epochs.

### F.2 MAIN EXPERIMENTAL SETTINGS AND ADDITIONAL RESULTS

To validate our approach across different domains, we conducted experiments on standard image classification and language modeling benchmarks. For image classification, we utilized CIFAR-10 and CIFAR-100, along with their corrupted variants CIFAR-10-C and CIFAR-100-C. These datasets provide a robust testbed for assessing model performance under various types of corruption, including noise, blur, weather effects, and digital transformations. The corruption types include Gaussian noise, shot noise, impulse noise, defocus blur, glass blur, motion blur, zoom blur, snow, frost, fog, brightness, contrast, elastic transformations, pixelation, and JPEG compression, representing a diverse set of real-world image degradations.

---

[1] https://scikit-learn.org/dev/auto_examples/neural_networks/plot_mlp_alpha.html

For language modeling experiments, we employed the WikiText-103 dataset, which contains over 100 million tokens extracted from high-quality Wikipedia articles. This dataset presents a challenging benchmark for evaluating the generalization capabilities of language models, with a diverse vocabulary and complex linguistic structures representative of real-world text.

All experiments were repeated three times with different random seeds (0, 1, and 2) to ensure the reliability of our findings. Our experimental infrastructure consisted of 2 Intel(R) Xeon(R) Gold 6226R CPUs @ 2.90GHz and 4 NVIDIA RTX 4090 24GB GPUs.

Table 8: Model Configurations

| Configuration | ViT-T (Tiny) | ViT-S (Small) | GPT-2 |
|---|---|---|---|
| Number of layers | 12 | 12 | 12 |
| Embedding dimension | 192 | 384 | 768 |
| Number of heads | 3 | 6 | 12 |
| MLP dimension | 768 | 1536 | 3072 |
| Patch size | 16 | 16 | - |
| Block size | - | - | 1024 |
| Number of parameters | 5.5M | 21.7M | 124M |

We implemented two Vision Transformer variants (ViT-T and ViT-S) for image classification tasks and a smaller modified version of GPT-2 (nanoGPT) for natural language processing tasks. Table 8 summarizes the architectural configurations of these models.

Table 9 summarizes the training hyperparameters used across our experiments. For Vision Transformers, we employed the AdamW optimizer with a base learning rate of 0.00025 and a cosine decay schedule over 200 epochs. We utilized a batch size of 256 and applied standard data augmentation techniques (Hendrycks et al., 2021; Mao et al., 2022), including MixUp (Zhang et al., 2018) and CutMix (Yun et al., 2019) to improve generalization. For GPT-2 (Radford et al., 2019), we used the AdamWSchedulefree optimizer (Defazio et al., 2024) with a constant learning rate of 0.001 for 20 epochs and a smaller batch size of 8 due to memory constraints.

Table 9: Training Configurations

| Hyperparameter | Vision Transformers | GPT-2 |
|---|---|---|
| Batch size | 256 | 8 |
| Optimizer | AdamW | AdamWSchedulefree (Defazio et al., 2024) |
| Base learning rate | 0.00025 | 0.001 |
| Learning rate schedule | Cosine decay | - |
| Training epochs | 200 | 20 |
| Weight decay | 0.05 | 0.05 |
| Dropout rate | 0.1 | 0.0 |
| MixUp $\alpha$, prob. | 0.8, 0.5 | - |
| CutMix $\alpha$, prob. | 1.0, 0.5 | - |
| $\lambda$ of $\mathcal{L}_{DB}$ | 0.01 | {0.5, 1.0, 5.0} |

The dual Banach regularization strength ($\lambda$) was set to 0.01 for Vision Transformer experiments, while for GPT-2, we explored a range of values (0.5, 1.0, and 5.0) to assess the impact of different regularization intensities on language modeling performance. All models were trained with a weight decay of 0.05 to prevent overfitting, with dropout applied only to the Vision Transformers.

Table 10 is the full experimental version of Table 1. $\mathcal{L}_{DB}$ consistently improves model performance under various corruption settings. Specifically, for CIFAR-10, the average accuracy across all corruptions increases from 87.15% to 89.53% (+2.38%) for ViT-T. Similarly, ViT-S exhibits an average accuracy improvement from 88.47% to 91.55% (+3.08%) with the inclusion of $\mathcal{L}_{DB}$. For noise corruptions such as Gaussian Noise and Shot Noise, it shows high robustness to severe input perturbations. These results demonstrate that $\mathcal{L}_{DB}$ effectively regularizes model training to better generalize under distributional shifts. We can observe similar results for CIFAR-100.

Table 10: Full experimental results with ViT-T and ViT-S on the CIFAR datasets.

**CIFAR-10 and CIFAR-10-C.**

| Models | ViT-T | | | ViT-T+$\mathcal{L}_{DB}$ | | | ViT-S | | | ViT-S+$\mathcal{L}_{DB}$ | | |
|---|---|---|---|---|---|---|---|---|---|---|---|---|
| Seeds | 0 | 1 | 2 | 0 | 1 | 2 | 0 | 1 | 2 | 0 | 1 | 2 |
| No Corruption | 94.05 | 94.08 | 93.9 | 95.78 | 95.66 | 95.79 | 94.69 | 95.05 | 95.37 | 96.74 | 96.76 | 96.89 |
| Gaussian Noise | 78.88 | 76.05 | 78.58 | 82.00 | 81.81 | 81.47 | 79.70 | 78.98 | 80.76 | 85.30 | 85.55 | 85.29 |
| Shot Noise | 82.40 | 80.65 | 82.20 | 85.34 | 85.50 | 85.1 | 83.53 | 83.09 | 84.42 | 88.45 | 88.67 | 88.42 |
| Impulse Noise | 86.80 | 86.35 | 86.36 | 88.33 | 88.98 | 88.85 | 87.75 | 87.96 | 89.06 | 91.77 | 91.08 | 91.56 |
| Defocus Blur | 91.07 | 91.27 | 91.20 | 93.02 | 92.71 | 92.94 | 91.79 | 92.02 | 92.61 | 94.24 | 94.11 | 94.32 |
| Glass Blur | 79.48 | 79.58 | 78.56 | 81.02 | 81.30 | 81.24 | 79.43 | 80.03 | 80.40 | 84.10 | 83.31 | 83.92 |
| Motion Blur | 88.24 | 88.58 | 88.76 | 90.80 | 90.56 | 90.88 | 89.01 | 89.83 | 90.32 | 91.95 | 92.00 | 92.51 |
| Zoom Blur | 90.95 | 91.28 | 91.32 | 92.98 | 92.96 | 92.84 | 91.71 | 92.34 | 92.82 | 93.98 | 94.39 | 94.64 |
| Snow | 89.77 | 89.71 | 89.61 | 92.05 | 92.14 | 91.96 | 90.80 | 90.86 | 91.15 | 93.95 | 93.83 | 93.82 |
| Frost | 90.29 | 90.22 | 90.48 | 92.74 | 92.57 | 92.55 | 91.33 | 91.68 | 92.04 | 94.37 | 94.40 | 94.36 |
| Fog | 87.54 | 88.01 | 87.62 | 91.33 | 91.18 | 91.37 | 89.43 | 89.90 | 90.77 | 92.97 | 92.91 | 93.16 |
| Brightness | 93.20 | 93.14 | 92.89 | 94.85 | 95.08 | 95.08 | 93.95 | 93.96 | 94.53 | 96.14 | 96.20 | 96.29 |
| Contrast | 89.40 | 90.31 | 89.41 | 92.13 | 92.19 | 92.53 | 90.57 | 91.24 | 92.22 | 94.02 | 93.57 | 94.33 |
| Elastic | 89.87 | 90.19 | 90.19 | 91.72 | 91.93 | 91.99 | 90.59 | 91.00 | 91.70 | 93.02 | 93.13 | 93.25 |
| Pixelate | 85.59 | 86.23 | 86.63 | 87.74 | 88.47 | 88.59 | 86.06 | 86.14 | 87.47 | 90.73 | 91.18 | 90.72 |
| JPEG | 84.81 | 84.01 | 84.21 | 85.98 | 86.06 | 85.98 | 85.74 | 85.51 | 85.64 | 88.11 | 87.99 | 87.87 |
| Avg. | 87.22 | 87.04 | 87.20 | 89.47 | 89.56 | 89.56 | 88.09 | 88.30 | 89.06 | 91.54 | 91.49 | 91.63 |

**CIFAR-100 and CIFAR-100-C.**

| Models | ViT-T | | | ViT-T+$\mathcal{L}_{DB}$ | | | ViT-S | | | ViT-S+$\mathcal{L}_{DB}$ | | |
|---|---|---|---|---|---|---|---|---|---|---|---|---|
| Seeds | 0 | 1 | 2 | 0 | 1 | 2 | 0 | 1 | 2 | 0 | 1 | 2 |
| No Corruption | 72.88 | 73.12 | 72.8 | 77.14 | 77.37 | 76.5 | 74.33 | 73.33 | 74.44 | 79.05 | 78.78 | 78.73 |
| Gaussian Noise | 49.24 | 48.82 | 49.05 | 55.09 | 53.30 | 52.87 | 49.25 | 48.26 | 50.74 | 57.43 | 58.29 | 57.64 |
| Shot Noise | 54.71 | 54.30 | 54.15 | 59.66 | 58.61 | 58.02 | 54.79 | 53.66 | 55.92 | 62.6 | 63.47 | 62.68 |
| Impulse Noise | 63.87 | 63.48 | 63.98 | 68.35 | 67.99 | 67.05 | 64.54 | 64.58 | 65.95 | 68.72 | 69.68 | 69.41 |
| Defocus Blur | 67.91 | 68.16 | 68.19 | 71.85 | 72.14 | 71.21 | 68.92 | 68.09 | 68.83 | 73.54 | 73.51 | 73.58 |
| Glass Blur | 47.58 | 45.44 | 46.49 | 48.98 | 48.92 | 49.6 | 47.01 | 47.00 | 48.03 | 50.77 | 51.27 | 49.22 |
| Motion Blur | 64.01 | 63.92 | 64.42 | 68.14 | 68.24 | 67.35 | 65.07 | 64.07 | 65.21 | 70.19 | 69.99 | 69.9 |
| Zoom Blur | 67.52 | 67.86 | 68.14 | 71.44 | 71.73 | 71.03 | 68.43 | 67.32 | 68.27 | 73.24 | 73.51 | 73.15 |
| Snow | 64.57 | 64.46 | 64.02 | 68.75 | 69.50 | 68.01 | 65.77 | 64.53 | 65.93 | 71.22 | 71.72 | 71.74 |
| Frost | 65.38 | 65.53 | 65.71 | 70.26 | 70.14 | 69.14 | 67.36 | 65.67 | 66.81 | 72.90 | 72.78 | 72.92 |
| Fog | 63.20 | 62.59 | 62.56 | 67.78 | 68.09 | 66.65 | 65.39 | 63.14 | 64.77 | 70.25 | 70.45 | 70.34 |
| Brightness | 70.46 | 70.75 | 70.44 | 74.99 | 75.23 | 74.27 | 72.03 | 70.78 | 72.00 | 77.04 | 76.96 | 77.30 |
| Contrast | 63.97 | 64.07 | 64.17 | 70.01 | 69.87 | 68.99 | 66.50 | 65.16 | 66.47 | 73.37 | 73.63 | 73.01 |
| Elastic | 65.82 | 66.21 | 66.20 | 69.82 | 70.10 | 69.19 | 66.66 | 65.86 | 66.81 | 71.18 | 71.31 | 71.20 |
| Pixelate | 61.88 | 61.99 | 62.00 | 68.18 | 69.01 | 67.26 | 64.46 | 62.27 | 64.54 | 71.30 | 71.39 | 70.84 |
| JPEG | 56.72 | 55.87 | 56.02 | 60.06 | 59.55 | 58.93 | 56.93 | 55.42 | 56.93 | 61.36 | 61.56 | 61.89 |
| Avg. | 61.79 | 61.56 | 61.70 | 66.22 | 66.16 | 65.30 | 62.87 | 61.72 | 63.15 | 68.34 | 68.63 | 68.32 |

# G USE OF LARGE LANGUAGE MODELS

In the preparation of this paper, a Large Language Model (LLM) was utilized as a general-purpose writing-assistance tool. The role of the LLM was limited to improving the quality of the prose, including enhancing clarity, correcting grammatical errors, and refining sentence structure to ensure the manuscript was articulate and readable.

