# OpenReview forum: "Rethinking Transformer through Dual Banach Spaces"
_ICLR.cc/2026/Conference — ICLR 2026 Conference Withdrawn Submission_

### Official Review · Reviewer_a7Z9 · 2025-10-28

**Soundness:** 2
**Presentation:** 2
**Contribution:** 2
**Rating:** 2
**Confidence:** 3

**Summary:**

This paper introduces a novel theoretical framework for understanding Transformers through the lens of dual Banach spaces. The authors prove that the exponentiated query-key kernel in attention can be interpreted as a bilinear form on Banach spaces, demonstrate that attention operates as a dual space operator, and interpret feed-forward networks as correction mechanisms between dual and primal solutions. Based on this framework, they propose Dual Banach (DB) regularization, which adds a functional norm-based penalty term derived from their theoretical analysis. Experiments on CIFAR-10/100, CUB-200-2011, and WikiText-103 show consistent improvements in accuracy and robustness when applying DB regularization to Vision Transformers and GPT-2.

**Strengths:**

1. **Novel theoretical perspective:** The dual Banach space framework offers a mathematically rigorous alternative to conventional Euclidean/RKHS analyses of Transformers.

2. **Practical regularization technique:** The proposed DB regularization (Equation 13) is concrete, implementable, and shows consistent empirical improvements across multiple domains

**Weaknesses:**

1. **Formatting issues:** The submission format violates conference guidelines-the margins are excessively wide and the header is missing. This is a serious issue and must be corrected for compliance.

2. **Limited experimental scale:** The experiments use relatively small models (ViT-Tiny/Small, GPT-2 Small) and modest datasets. It remains unclear whether the proposed regularization scales effectively to larger architectures and datasets.

3. **Lack of comparison to Transformer variants:** The analysis and experiments omit discussion or evaluation against other attention variants (e.g., sparse, linear, or flash attention). Including encoder-decoder architectures would further validate generality.

4. **Unverified FFN interpretation:** The claim that FFNs act as corrector modules between dual and primal spaces is intriguing but empirically underexplored. Extending the ℓ₁-SVM validation to actual trained FFN layers in Transformers would strengthen this argument.

**Questions:**

See weaknesses

---

### Official Review · Reviewer_RNcz · 2025-10-31

**Soundness:** 2
**Presentation:** 3
**Contribution:** 2
**Rating:** 4
**Confidence:** 3

**Summary:**

This paper presents a novel theoretical framework for interpreting Transformer architectures via dual Banach spaces. The authors interpret that the self-attention mechanism corresponds to a dual space operator, while feed-forward networks serve as correction mechanisms bridging dual and primal solutions. Based on this perspective, the authors introduce Dual Banach regularization and demonstrate its effectiveness in enhancing robustness and generalization on image classification and language modeling benchmarks.

**Strengths:**

- This work propose a novel dual Banach space perspective on Transformers, explain attention as dual operator and FFN as primal-dual corrector, inspiring new designs/analyses.
- Experimental results show improved performance on image classification with both clean and corrupted image (CIFAR-10, 100, 10-C, 100-C, CUB-200) and language modeling (WikiText-103) tasks.

**Weaknesses:**

- The paper appears to use an incorrect format for the conference, with narrower left and right margins than standard, giving more space for content.
- The empirical validation in Appendix D demonstrates that an MLP can learn to act as a corrector only in an SVM setting. Could this be extended to show its role as a corrector in a trained Transformer or at least a simplified Transformer-like model?
- What are the effects of different choices of $\varphi$ in Eq. 5? Line 190 states that $\varphi$ needs only to be a strictly increasing smooth function on $\mathbb{R}^+$, so what happens when we set $\varphi(x)=x$ or other functions?
- The experimental section lacks evaluations on large-scale datasets, such as ImageNet and its corrupted variants (-C, -P, etc.).
- The paper does not sufficiently clarify the distinct mechanisms by which weight decay and $\mathcal{L}_{DB}$ reduce overfitting, despite stating their complementarity. The empirical results in Table 4 are limited and do not convincingly support this claim due to a narrow hyperparameter search. A wider exploration of $\lambda$ and weight decay values is necessary to better demonstrate their combined effect. In particular, the GPT-2 experiments should show that optimizing both hyperparameters (ie, showing the perplexity minima) outperforms using either one alone to strengthen the argument.
- Can this framework be applied to other models or architectures, such as MLPs, SSMs, etc.?

**Questions:**

Please refer to weaknesses.

---

### Official Review · Reviewer_bMrT · 2025-11-01

**Soundness:** 3
**Presentation:** 2
**Contribution:** 3
**Rating:** 6
**Confidence:** 3

**Summary:**

This paper proposes a new theoretical framework for Transformers, interpreting them through the lens of dual Banach spaces. The authors argue that self-attention is interpreted as an operator to find a solution in a dual space, while the FFN acts as a learned primal-space correction. This motivates a single-layer norm bound that is turned into a practical Dual Banach (DB) Regularization. The authors demonstrate empirically that this regularizer improves generalization, model robustness on corrupted inputs, and perplexity across vision and language tasks.

**Strengths:**

1. The paper presents a novel and mathematically principled perspective on the Transformer architecture. The central hypothesis that the Attention/FFN split directly corresponds to a dual-solution/primal-correction step is an elegant theoretical explanation.
2. From this theory, the authors introduces practical application. The derived Dual Banach (DB) regularization is shown to be effective, complementary to existing method. The empirical validation is well-rounded, covering synthetic data, vision tasks, and language tasks. The results show clear improvements in robustness.
3. The validation experiment in Appendix D to test if an FFN can learn a proximity correction step, is a good supporting evidence for the paper's core hypothesis.

**Weaknesses:**

1. The paper's motivation for introducing a new dual Banach space theory is weak. It mentions some alternative mathematical lenses [1, 2, 3], but fails to articulate why these are insufficient. Furthermore, it omits discussion of the relevant body of work on kernel-based approximations of attention [4]. Without this comparison, the necessity for this new framework is not well-established.
2. The interpretation of the FFN as a "correction mechanism"  is presented as a hypothesis, not a formal proof. While supported by an appendix experiment, this link remains an interpretation rather than a direct mathematical derivation from the core theory.
3. The claim that dual Banach regularization is complementary to weight decay is not fully substantiated. The results in Table 4 show that simply increasing the weight decay strength (from 0.0 to 0.10) also improves the baseline performance. It is plausible the gains from dual Banach regularization could be matched by simply tuning the weight decay hyperparameter more extensively, which undermines its necessity.

**Questions:**

Q1. Could the authors better situate their contribution? The paper's motivation  would be much clearer with a detailed discussion of how this dual Banach space framework compares to, and improves upon, existing mathematical interpretations, including the kernel-based views [1]?

Q2. It appears to me that the non-linear activation function (e.g., GeLU/ReLU) within the FFN is not considered in the analysis of the bound in Proposition 4.1. Can the authors discuss how to adjust this analysis to this more practical setting of the Transformer? I would appreciate any general directions from the author on this matter and not the full analysis in the rebuttal.

Q3. This is likely a minor point, but could the authors clarify the paper's claims regarding its theoretical scope? The theory is motivated by moving beyond standard RKHS analysis , yet the proof (Theorem 3.1) appears to rely on a specific construction from Hilbert spaces . A more explicit clarification on whether the claims hold generally or for this specific construction would be helpful.

[1] Tsai et al. Transformer dissection: An unified understanding for transformer's attention via the lens of kernel. EMNLP 2019.

[2] Geshkovski et al. A mathematical perspective on transformers. 2023.

[3] Elhage et al. A mathematical framework for transformer circuits. 2021.

[4] Choromanski et al. Rethinking Attention with Performers. ICLR 2021.

---

### Official Review · Reviewer_4L8p · 2025-11-01

**Soundness:** 2
**Presentation:** 2
**Contribution:** 2
**Rating:** 2
**Confidence:** 4

**Summary:**

The paper proposes a functional-analytic view of Transformers by modeling attention within dual Banach spaces, showing the query–key kernel is a bilinear form and interpreting attention as a dual-space operator. It argues the feed-forward network acts as a learned correction from the dual representation back toward a primal solution, offering a principled view for the Transformer block. Building on this, the authors introduce “Dual Banach” regularization and report improved generalization and smoother decision boundaries across vision and language benchmarks.

**Strengths:**

- The connection of Transformer architecture and the functional view is well motivated and the work also have a empirical verification for the MLP layer as correction mechanisim, furthermore support their functional view.
- The proposed norm-bound modification to the loss function is well motivated by the theoretical framework. Which demonstrates some improvements in CIFAR-10/100 and Wikitext-103 tasks considered.

**Weaknesses:**

- The paper appears to use an incorrect template, as the header “Under review as a conference paper at ICLR 2026” does not appear on the pages.
- The experimental scope is limited, with no large-scale benchmarks such as ImageNet classification, which weakens empirical support for the framework.
- Language-model gains are small, with less than a 1.0 perplexity reduction across all $\lambda$ settings; combined with the absence of large-scale tasks, this highlights the narrow empirical scope.

**Questions:**

- Would the authors consider adding a large-scale image classification task (e.g., ImageNet) or other fine-tuning benchmarks to strengthen the evaluation?
- For WikiText-103, performance appears to improve as $\lambda$ increases. Could the authors include a hyperparameter study to identify the performance-optimal $\lambda$ and report sensitivity curves?

---

### Note · Authors · 2025-11-22

**Comment:**

We sincerely thank the reviewers for their detailed comments and helpful suggestions.

After carefully considering the feedback, particularly regarding the presentation and experimental scope, we have decided to withdraw our submission at this time. We plan to revise the paper to address the reviewers' concerns and improve the theoretical and empirical solidity of our work.

**Withdrawal Confirmation:**

I have read and agree with the venue's withdrawal policy on behalf of myself and my co-authors.